# Dendrimer Nanodevices and Gallic Acid as Novel Strategies to Fight Chemoresistance in Neuroblastoma Cells

**DOI:** 10.3390/nano10061243

**Published:** 2020-06-26

**Authors:** Silvana Alfei, Barbara Marengo, Guendalina Zuccari, Federica Turrini, Cinzia Domenicotti

**Affiliations:** 1Department of Pharmacy (DiFAR), University of Genoa, Viale Cembrano, 16148 Genoa, Italy; zuccari@difar.unige.it (G.Z.); turrini@difar.unige.it (F.T.); 2Department of Experimental Medicine—DIMES, University of Genoa, Via Alberti L.B., 16132 Genoa, Italy; barbara.marengo@unige.it (B.M.); cinzia.domenicotti@unige.it (C.D.)

**Keywords:** Human neuroblastoma, gallic acid (GA), ROS-mediated anticancer effect, polyester dendrimers, dendrimer nanoformulations

## Abstract

Human neuroblastoma (NB), a pediatric tumor inclined to relapse, after an initial response to therapy, usually develops resistance. Since several chemotherapeutics exert anticancer effect by increasing reactive oxygen species (ROS), NB cells overproduce antioxidant compounds becoming drugs-resistant. A strategy to sensitize NB cells to chemotherapy involves reducing their antioxidant defenses and inducing ROS overproduction. Concerning this, although affected by several issues that limit their clinical application, antioxidant/pro-oxidant polyphenols, such as gallic acid (GA), showed pro-oxidant anti-cancer effects and low toxicity for healthy cells, in several kind of tumors, not including NB. Herein, for the first time, free GA, two GA-dendrimers, and the dendrimer adopted as GA reservoir were tested on both sensitive and chemoresistant NB cells. The dendrimer device, administered at the dose previously found active versus sensitive NB cells, induced ROS-mediated death also in chemoresistant cells. Free GA proved a dose-dependent ROS-mediated cytotoxicity on both cell populations. Intriguingly, when administered in dendrimer formulations at a dose not cytotoxic for NB cells, GA nullified any pro-oxidant activity of dendrimer. Unfortunately, due to GA, nanoformulations were inactive on NB cells, but GA resized in nanoparticles showed considerable ability in counteracting, at low dose, ROS production and oxidative stress, herein induced by the dendrimer.

## 1. Introduction

In the last decades, the correlation between oxidative stress (OS) and cancer onset has been highlighted by several pieces of evidence, but with contradictory findings. In fact, during the neoplastic transformation, there is an improvement in the production of reactive oxygen species (ROS), which supports tumor cell growth, by stimulating both redox-modulated signal transduction routes and transcription agents. In this regard, the use of ROS scavenging synthetic or natural compounds, counteracting the activation of redox sensitive pathways, could be employed as anti-carcinogenic for preventing the tumors onset and for treating the disease in the early stages.

When chemotherapeutic drugs induce an increase in intracellular ROS levels, the intrinsic antioxidant defense decreases and cancer cell death occurs. Therefore, in this perspective, antioxidants, contrasting the cytotoxic role of ROS, could favor neoplastic progression. 

In this regard, an effective anti-tumor strategy could be to combine drugs capable of increasing ROS production with compounds able to reduce the intracellular antioxidant defenses. Human neuroblastoma (NB) is a solid tumor affecting young children, responsible for 15% of childhood cancer mortality [1,2,3]. 

NB is a heterogeneous tumor that origins from neural crest elements of the sympathetic nervous system and includes both low-risk forms, capable of regressing spontaneously or of differentiating into benign ganglioneuroblastoma, and high-risk (HR) forms, featured by metastatic disease and/or presence of MYCN proto-oncogene amplification. The amplification of MYCN is a biomarker still used today for early prognosticating risk and it is an indicative factor of poor prognosis [4]. 

Currently, HR patients undergo treatments, which include intensive and toxic chemotherapy followed by surgical resection, myeloablation and rescue of autologous stem cell, radiotherapy, and intensive immunotherapy [5]. 

Chemotherapeutic standard treatments consist of multi-drugs therapies, which combine several compounds including doxorubicin and etoposide (ETO). In particular, ETO is widely used [6,7,8] and induces a ROS-mediated anti-tumor effect. Unfortunately, its severe side effects [9,10] and chemoresistance limit its clinical success [11]. 

Chemoresistance is a multifactorial phenomenon and, recently, it has been demonstrated that ETO-resistant NB cells had high levels of glutathione (GSH) [12], which is the most important intracellular antioxidant thiol crucially involved in both cancer progression and chemoresistance [13]. In this context, the current trend of research, in the field of cancer treatment, increasingly focuses on developing alternative preventive and/or therapeutic strategies, based on the use of less toxic natural bioactive compounds, such as polyphenols, having both pro-oxidant and antioxidant activities, usually correlated to the dose [14]. 

Polyphenols, due to their antioxidant properties, have the possibility to act as preventive anti-cancer compounds [15], and, thanks to their pro-oxidant effects, they can work as mimics of chemotherapeutic drugs, inducing ROS-mediated cancer cell death [16].

It is the case of gallic acid (GA, Figure 1), which is the 3,4,5-*tri*-hydroxyls derivative of benzoic acid and represents one of the major phenolic acids present in various edible natural products, such as green tea, gallnuts, oak bark, apple peels, grapes, strawberries, pineapples, bananas, and many other fruits [17].

The ingestion of these foods is justifiably correlated to beneficial outcomes for the human health, such as reduced risk of cardiovascular diseases and myocardial infarction, reduced predisposition to tumor diseases, and improvement of the quality of life in people suffering from neurodegenerative diseases or early stage tumors. Inside plants, GA is one of the secondary metabolites involved in the formation of the galatotannin-hydrolysable tannins, but, in the biomedical sector, it has long attracted the interest of scientists for its ambivalent antioxidant/pro-oxidant behavior [18] and its capacity in counteracting diseases correlated to OS, through its anti-bacterial, anti-viral, anti-inflammatory, anti-neurodegenerative, and anticancer activities [15,16,17,19,20,21,22,23,24,25,26,27,28,29]. 

Furthermore, GA finds applications in several other areas, including food and cosmetic industry, as natural preservative in food, beverages, beauty products, and essential oils, because of its free radicals scavenging activity (RSA) [30].

Several studies report that GA can counteract cancer growth and progression, thanks to its anti-invasive and anti-metastatic activities and that it could be useful both to treat neoplasia, inducing ROS production such as ETO, and to prevent malignant transformation, exerting a protective antioxidant effect on healthy cells [31,32,33,34]. 

Based on these findings and considering that, to our knowledge, no evidence proving significant GA activity against human NB cells has yet been reported yet, this compound, for the first time, was investigated as novel alternative therapeutic approach to treat this pediatric tumor. Unfortunately, GA is sparingly soluble in water and alcohols [35], practically insoluble in hydrophobic solvents, and its activity and clinical applications are hampered by additional drawbacks such as high instability and low bioavailability. 

Luckily, the recent advances in the field of nanomedicine and in the use of nanoparticles (NPs), as convenient drug carrier systems capable of improving drug solubility, half-life, and bioavailability and of lessening the metabolism and systemic toxicity of several problematic bioactive compounds, including polyphenols, increasingly allow for further expansion of the possibilities of anticancer treatments by employing natural compounds. 

In the last decades, dendrimers have arisen as the most talented NP carrier systems endowed with the possibility of revolutionizing cancer treatments. Dendrimers can be employed to efficiently deliver anti-neoplastic drugs and typically are used as scaffolding with a well-defined architecture or as nanovehicles to conjugate, complex, or entrap therapeutic drugs.

Structurally, dendrimers are symmetric monodisperse [36] tree-like macromolecules, with both internal cavities for guest molecule encapsulation [37] and several peripheral chemical groups for further functionalization by covalent bond.

Interestingly, they possess an unusually low intrinsic viscosity that makes easy their transport in the blood [38,39]. 

They possess the capability of controlling molecular weight, hydrophilicity, solubility [39,40,41], bioavailability, and pharmacokinetic behavior of transported drugs. 

Thanks to dendrimer’s ability in establishing strong interactions with several drugs, their loading results facilitated and their systemic toxicity, often due to the initial massive drug release (burst release), is minimized [39,40,41]. 

Among dendrimers, positively charged and commercially available poly(amidoamine) (PAMAMs) and polypropyleneimine (PPIs) are well-functioning for several biomedical applications, including gene therapy and drug delivery, but their clinical application is hampered by high non-selective cytotoxicity, hemolytic toxicity, genotoxicity, low biodegradability, fast removal from circulatory system, and high uptake in the reticuloendothelial system [42]. 

On the contrary, uncharged polyester dendrimer scaffolds appear more attractive and suitable for biomedical applications, because respectful of physiologic membranes and characterized by a good biodegradability [36,43,44].

Concerning GA-enriched dendrimers, PAMAM dendrimers have been adopted both for covalently linking and for encapsulating GA [45,46], while PPI dendrimers have been used to solubilize GA and to control its release profile [47]. 

In addition, examples of biodegradable polyester-based dendrimers in which GA is either the repeated monomeric unity [48,49] or makes part of the dendrimer backbone [50] are reported. 

In this work, thinking about a future clinical application of GA, to ameliorate its solubility and stability, to slow down its metabolism, and to minimize the active dosage, a lab-made biodegradable dendrimer (namely, **4**) [51,52,53,54,55,56] was adopted for formulating GA in NPs (Figure 2).

In addition, since it has been recently demonstrated that dendrimer **4** possesses a ROS-mediated anticancer activity, at least on NB cells sensitive to ETO comparable to that of the drug [57], it was selected as nanocarrier for GA, with the aim at inducing high mortality in human NB cells by improving GA cytotoxic efficacy, at lower doses. 

Furthermore, it has been reported [58] that the internal construction of a dendrimer, depending on its chemical structure, is suitable for complexing both hydrophobic and hydrophilic compounds [59]. 

Hence, in addition to being chosen for its intrinsic cytotoxic activity, the polyester-based dendrimer **4**, having hydrophilic characteristics, was considered suitable to host a hydrophilic molecule such as GA, with which it can establish hydrogen bonds. 

In addition, being uncharged, it has been thought to also be well compatible with the phenyl hydrophobic portion of GA. Moreover, dendrimer **4** was considered a proper candidate carrier to entrap GA because, thanks to its high generation, it could offer more space to host drugs, thus allowing a higher loading [60]. 

Furthermore, the large chemical architectures, with wide surface and high molecular weight, achievable with high generation dendrimers such as **4**, characteristically remain in the blood circle for longer periods [61].

Finally, dendrimer **4** was preferred because its uncharged polyester-based hydrolysable architecture matches the requirements of low toxicity and high biodegradability desirable for biomedical applications [36,43,44]. 

Starting from these assumptions, using dendrimer **4**, firstly a GA-enriched dendrimer (GAD **6**, Figure 3a), in which GA is covalently linked on the dendrimer surface by ester type bonds, was synthetized according to a reported procedure [54]. 

Secondly, a GA-loaded dendrimer (GALD **7**, Figure 3b), in which GA is physically complexed with dendrimer **4**, through both inside entrapment and surface absorption, was prepared and totally characterized.

After investigations concerning the dose- and time-dependent GA effects on two NB cell lines differently sensitive to ETO [12,62], the biological activities of dendrimer **4**, free GA, and GA-enriched dendrimers **6** and **7** at specific selected doses were investigated as well. 

## 2. Material and Methods

### 2.1. Chemicals and Instruments

All chemical products, including gallic acid (GA) and solvents, were purchased from Merck (formerly Sigma-Aldrich, Darmstadt, Germany). Copies of FTIR (lab-made) and ^1^H and ^13^C NMR spectra (database of Aldrich, Darmstad, Germany) of commercial GA are accessible in Appendix A. Synthetized dendron intermediates (D4BnA, D4BnOH, D5BnA, and D5ACOOH) necessary to achieve dendrimer **4**, adopted as scaffold-carrier to link and entrap GA (**1**), were prepared according to what reported previously [51,52,53]. Their chemical structures are observable in Appendix A. Dendrimer **4** and GA-enriched dendrimer GAD **6** were prepared according to procedures previously reported [54,55,56].

Characterization data of dendrimer **4** and GAD **6**, counting copies of FTIR, ^1^H and ^13^C NMR spectra are accessible in SM (Appendix A, Appendix A, and Appendix A). The protected/activated GA derivative (GA-TBDMS-Cl) necessary to esterify peripheral hydroxyls of dendrimer **4** was prepared reproducing the synthetic pathway shown in SM (Appendix A, Appendix A) [54].

Reagents were no furtherly purified, while solvents were dried and distilled according to standard procedures. The fraction of petroleum ether with boiling point 40–60 °C was used. Methanol used for UV analysis was of HPLC grade and was purchased from Merck (formerly Sigma-Aldrich, Darmstadt, Germany). Triple distilled water (TDW) was used for PBS and water media preparation and was purchased from Distilled Water Supplies, Podington NN29 7XA Northants, UK. PBS and TDW were purified by filtration through 0.22 µ filters (Millipore Sigma Life Science Center, Burlington, MA, USA). Melting points, FTIR, and ^1^H and ^13^C NMR spectra were acquired on the same instruments and with the same modalities previously described [57].

Centrifugations, freeze-drying, Dynamic Light Scattering (DLS), and Z-potential determinations were performed on the same instruments and with the same modalities previously described [57]. UV–Vis determinations were achieved on an Agilent UV-Visible spectrophotometer Cary 100 (Varian Co., Santa Clara, CA, USA) with 0.5 nm resolution. Samples solutions were analyzed in quartz cells of 10-mm path length. Scanning electron microscopy (SEM) images (Appendix A) were acquired on a Leo Stereoscan 440 instrument (LEO Electron Microscopy Inc., Thornwood, NY, USA). Aluminium-backed silica gel plates (Merck DC-Alufolien Kieselgel 60 F254, Merck, Washington, DC, USA) were used to perform thin layer chromatography (TLC) and spots detection was allowed by UV light. Elemental analyses were determined as previously described [57]. Anhydrous magnesium sulfate was employed to dry organic solutions, which were evaporated using a rotatory evaporator operating at reduced pressure of 10–20 mmHg (BUCHI Italia s.r.l, Cornaredo, Italy). 

### 2.2. Complexation Reaction of GA with Dendrimer **4** (GALD **7**)

A test tube was flamed under nitrogen gas and was used as the reactor in which GA (100.0 mg; 0.588 mmol) was dissolved in 3.2 mL of methanol (MeOH), achieving a GA solution 31.3 mg/mL. The solvent was evaporated in vacuum and the solid residue was dried at reduced pressure. A solution of **4** (100.5 mg, 0.01381 mmol) in MeOH (24.8 mL) was prepared with concentration 4 mg/mL and was used to re-dissolve the solid residue of GA. The novel solution, containing both dendrimer **4** and GA, was kept under vigorous stirring, in the dark, for 48 h, and no precipitate was formed during stirring. The clear pale-yellow solution was partially evaporated at reduced pressure until a white solid (free GA) began to settle on the walls of the flask. Then, the evaporation was stopped and the crystallization of the non-complexed GA was completed by placing the turbid solution at 4 °C overnight. The solid crystallized, initially supposed to be free GA, was recovered by centrifugation at 3360 rpm, washed with diethyl ether (Et_2_O), brought to constant weight at vacuum (10.6 mg) and analyzed by FTIR technique that instead confirmed the structure of not complexed **4**. The solution containing the GA-loaded dendrimer was finally evaporated, washed with Et_2_O and brought to constant weight at reduced pressure (188.2 mg). The GA-loaded dendrimer **7** (GALD) was obtained as off-white glassy solid that was stored in a dryer on P_2_O_5_. 

FTIR (KBr, cm^−1^): 3600–3200 (large band OH of **4**), 3497, 3383, 3286, (different OH of GA), 2932 (**4**), 1734 (C=OO of **4**), 1688 (GA), 1615 (GA), 1425, 1320, 1028, 867, 734, 559 (bands of GA). ^1^H NMR (300 MHz, DMSO-*d6*), δ (ppm): 1.01, 1.16, 1.18, 1.23, 1.34 (five s signals, 186H, CH_3_ of generations), 1.70 (m, 2H, CH_2_ propandiol), 3.52 (dd, 128H, CH_2_OH and 2H, CH_2_O propandiol, overlapped signal), 3.95–4.20 (m, 120H, CH_2_O of four generations and 2H, CH_2_O propandiol, overlapped signal), 4.30 (br s, 64H, OH), 7.32 (s, GA phenyl CH=).

### 2.3. FeCl_3_ Essay 

Approximately 1 mL of 5% ethanol solution of GALD **7**, pale yellow colored, was inserted in a test tube and was treated with 2–3 drops of 5% FeCl_3_ freshly prepared. For comparison purposes, 1 mL of 5% ethanol solution of GA **1** was assayed in the same conditions. The new coloration of the two solutions were compared. 

### 2.4. Chemometric Analysis of FTIR Spectral Data by Principal Components Analysis 

KBr pellets containing a sample of GA, dendrimer **4** and GALD **7** were prepared and were used to acquire the FTIR spectra of each compound in triplicates. The spectral data of the nine spectra achieved were processed with Principal Component Analysis (PCA) by using Analyse-it^®^ for Microsoft excel statistical software. 

### 2.5. UV–Vis Spectrophotometric Analysis of GALD

The total UV–Visible spectrum of GALD was acquired in methanol solution. Particularly, a solution of GALD (15.9 mg) in MeOH (50 mL) was diluted (1:20) with further methanol in order to avoid the signal saturation. The total spectrum was acquired in duplicate at room temperature (r.t.) (25 ± 1 °C), counter to a blank solution (methanol).

### 2.6. Determination of GA Content in GALD **7**: Drug Loading Percentage (DL%) 

The actual amount of GA complexed with the dendrimer **4**, representing its loading capacity, was assessed by the common and well-known procedure that makes uses of Folin–Ciocalteu reagent, the UV–Vis spectroscopy and the standard GA calibration curve [63]. 

#### 2.6.1. Standard GA Calibration Curve for Folin–Ciocalteu Method

To build the standard GA calibration curve necessary to determine GALD drug loading, stock solutions at GA concentrations of 10, 20, 25, 40, and 50 µg/mL in methanol were prepared. For each concentration, aliquots of 0.2 mL were added with the Folin–Ciocalteu reagent, diluted 1:10 with Milli-Q water (1 mL) and aqueous sodium carbonate 7.5% *w/v* solution (0.8 mL) and were stirred. The mixture was kept in the dark at r.t. (25 ± 2 °C) for 30 min. After this reaction period, the absorbance was measured at 760 nm counter to a blank solution not containing GA prepared with the same procedure [64]. 

From the stock solutions of GA (10–50 µg/mL), six separate series of solutions were prepared and analyzed achieving the average absorbance for each GA concentration ± standard deviation (SD). 

The values of average absorbance ± standard deviation (A_average_ ± SD) measured for each concentration of GA (C_GA_) are reported in Appendix A in Appendix A.

The statistical descriptive parameters concerning the calibration set are reported in Appendix A (Appendix A).

A_average_ and C_GA_ (µg/mL) data in Appendix A were used to work out the GA calibration curve by least squares (LS) method shown in Appendix A (Appendix A). 

The equation of the linear regression obtained (Equation (1)), showed a regression coefficient (R) of 0.9943 and a R^2^ value of 0.9887.
y = 0.011x + 0.0059(1)
where y is the absorbance (A) measured at λ = 760 nm and x is the GA concentration (C_GA_) (µg/mL).

The linearity and sensitivity of the developed calibration line were evaluated, by confirming the statistical significance of its slope, through the analysis of variance (ANOVA), performing both the Fischer test and the Student t_n−2_ test. Statistical significance was established at the *p*-value < 0.05.

By using Equation (1), the predicted GA concentrations (C_GAp_) were computed for each sample and are reported in Appendix A. 

Appendix A contains also the residuals and the absolute percentage errors (%w/v, mg/100 mL). The C_GA_s versus the C_GAp_s are reported in graph and the linear regression correlating the two sets of data is available in Appendix A (Appendix A). The value of R was 0.9943. 

A_average_ measured and GA µM concentrations (last column in Appendix A, Appendix A) were used to calculate the molar extinction coefficient (ε) of the oxidized form of GA (GAOx), formed for the action of Folin–Ciocalteu reagent, according to the simplified version (Equation (3)) of Lambert–Beer law (Equation (2)) usable when the optical length path is 1 cm.
A = *ε b M*(2)
where *b* is the length of the optical path and is equal to 1 cm and *M* is the GA molar concentration.
A = *ε M*(3)

Briefly, A_average_ values versus C_GA_ concentrations (µM) were plotted on a graph (Appendix A, Appendix A) and it was verified that the intercept was not significantly different from zero. Consequently, the linear regression equation with intercept zero was obtained (Equation (4)), the regression coefficient (R) was 0.9945, and R^2^ was 0.9887.
y = 0.0019x(4)
where y is the absorbance (A) measured at λ = 760 nm and x is the C_GA_ (µM). In a graph of A versus C, the slope is the molar extinction coefficient (ε) of the compound under study. In this regard, the molar extinction coefficient of GAOx (ε_GAOx_) was 0.0019 µM^−1^ L cm^−1^, i.e., 1900 M^−1^ L cm^−1^.

#### 2.6.2. Estimation of GALD **7** Drug Loading

First, 15.9 mg of GALD **7** were dissolved in MeOH (50 mL) to promote the release of GA in order to make feasible its determination. An aliquot of 0.1 mL was diluted to 10 mL with MeOH obtaining a concentration of GALD of 31.8 µg/mL. Six aliquots (0.2 mL) of this solution were mixed separately with the same reagents, as performed for constructing the calibration curve and kept in the dark. After 30 min, each aliquot was analyzed by UV–Vis spectrometer acquiring the absorbance at 760 nm counter to a blank solution of empty the dendrimer [64]. 

The measured absorbance (A), the related C_GA_ (µg/mL) computed by using Equation (1), and the molar extinction coefficients of GA released by the complex and oxidized by Folin–Ciocalteu reagent, (ε_GAOxC_) calculated by Equation (3) for each aliquot, are included in Appendix A.

The average absorbance ± SD (A _average_ ± SD) resulted to be 0.2650 ± 0.004 and the average GA concentration ± SD (C_GA average_ ± SD) in the sample of GALD 7 analyzed (31.8 µg/mL), resulted to be 23.55 ± 0.37 µg/mL. The drug loading (DL%) was calculated by Equation (5) and was of 74.1%:
(5)$$\mathrm{DL}\%=\frac{\mathrm{Weight of GA in GALD}}{\mathrm{Weight of GALD}}\;\times \;100$$

The molar extinction coefficient of GA released by the complex and oxidized by Folin–Ciocalteu reagent (ε_GAOxC_) at the obtained concentration of 23.6 µg/mL was computed according to the simplified version of Lambert–Beer law (Equation (3)) and was found to be 1913 M^−1^ L cm^−1^. 

Once the content of GA for a given weight of GALD **7** was known, it was possible to calculate the GA moles (moles _GA_) loaded per dendrimer **4** mole (mole **_4_**), and, from these data, the Molecular Weight (MW) of **7** was calculated. 

In this regard, MW of **7** was achieved according to a procedure previously reported in literature and accepted as valid [37,40,41,57]. In addition, the GA entrapment efficiency (EE%) [57], calculated by Equation (6), was found to be 148.4%:
(6)$$\mathrm{EE}\%=\frac{\mathrm{Weight of GA in GALD}}{\mathrm{Weight of GA fed initially}}\;\times \;100$$

### 2.7. In Vitro GA Release Profile from GALD 

To predict GALD stability in vivo and its half-life, in vitro GA release from dendrimer **7** was evaluated by performing the equilibrium dialysis technique at physiological conditions of blood at 37 °C and pH = 7.4. A sample of GALD of 5 mg (equivalent to 2.3 mg of GA) was dissolved in 10 mL phosphate buffer saline solution (PBS) and inserted in a membrane bag for dialysis (Himedia, MWCO, molecular mass cut off 12,000–14,000, pore size 2.4 nm). 

The membrane bag with the GALD solution was immersed in 300 mL PBS in a beaker and kept under magnetic stirring. Aliquots of 2 mL were withdrawn at fixed time points for 96 h and 2 mL of fresh PBS were added in the backer.

To prevent the complexation of GA in PBS, five fractions of each aliquot were firstly added of 0.1% *w/v* EDTA, and analyzed by UV spectrometer, for determining GA concentration, acquiring the relative absorbance at 261 nm. 

The absorbance (A) measurements were obtained against a PBS/0.1% EDTA *w/v* solution as a blank and were expressed as average absorbance ± standard deviation (A _average_ ± SD). To achieve the average GA concentration ± standard deviation (C_GA average_ ± SD) for each taking, a standard calibration curve of GA in PBS/0.1% EDTA *w/v* was prepared following a procedure similar to that described for Folin–Ciocalteu method and by measuring absorbance at λ max = 261 nm. The standard GA calibration equation obtained (Equation (7)) showed a R^2^ value of 0.9995.
y = 0.0943x + 0.0054(7)
where y is the absorbance measured and x is the relative GA concentration. 

The GA released at the time point of 1/2, 1, 2, 4, 8, 12, 24, 36, 48, 60, 72, 84 and 96 h was reported as cumulative GA release percentage (%).

### 2.8. Particle Size, Polydispersion Indexes (PDI) and Z-Potential of Dendrimer **4** and GALD 

The hydrodynamic size (diameter), PDI, and Z-potential (mV) of dendrimer **4** and GALD **7** particles were found using Dynamic Light Scattering (DLS) analysis, and were determined following a procedure previously described [57]. Measurements were performed in water mQ as medium at max concentration of the compounds under study of 3 mg/mL (pH = 7.4). Differently from the previous work [57], PDI value was reported as the mean of three measurements ± SD made by the instrument on the sample. 

### 2.9. Evaluation of GALD Solubility

The GALD solubility was evaluated in water and ethyl acetate, according to a method previously described for polyester-based dendrimer formulations and considered valid [37,57]. 

In this work, 6.3 mg of **7** exactly weighed, were added with a starting aliquot of 50 µL of solvent. Then, aliquots of 50 µL were added if necessary. The sample was reputed soluble when clear solutions stable in time were achieved.

### 2.10. Cell Culture Conditions and Treatments

HTLA-230 human stage-IV NB cells were kindly provided by Dr. L. Raffaghello (G. Gaslini Institute, Genoa, Italy). HTLA-ER were selected by HTLA-230 parental cells as previously reported [12]. To determine the biological actions of GA, time- and dose-dependent experiments were carried out, by treating both cell populations for 48 and 72 h, with increasing concentrations of GA (10–150 μM). 

Then, both cell populations were treated for 48 or 72 h, with 0.169 μM dendrimer **4** (the dose active to provide a ROS-mediated cytotoxic effect [57]), with 0.1656 GALD (the dose capable of providing 0.169 μM dendrimer **4**), with 21.20 μM GA (the dose of GA provided by GALD used), and with 0.3313 μM GAD (the dose capable of providing 21.20 μM GA). 

Stock solutions of the four compounds were prepared in DMSO employing final amounts of solvent unable to affect any cell responses analyzed.

### 2.11. Cell Viability Assay

Cell viability was determined as previously described [57].

### 2.12. Detection of Reactive Oxygen Species (ROS) Production

The production of ROS was evaluated by using 2′-7′ dichlorofluorescein-diacetate (DCFH-DA; Sigma) [65] and following a reported procedure [57]. 

### 2.13. Statistical Analyses

Data are expressed as means ± SD. Statistical significance of differences was determined by one-way analysis of variances (ANOVA). *p* < 0.05 was considered statistically significant.

## 3. Results and Discussion

### 3.1. Preparation of GA-Dendrimer Nanoformulations GAD (**6**) and GALD (**7**)

Dendrimer NPs have proved to be particularly efficient in improving drugs therapeutic efficacy at lower dosages by behaving as “excipients” or enhancers of permeability, by altering the barrier properties of the intestinal epithelium, thus enhancing the permeation ability of the transported drug [61]. 

Following this current and successful trend, GA was previously formulated in NPs, by its covalent bond to a fifth-generation, biodegradable dendrimer matrix (**4**) obtaining a peripherally GA-decorated dendrimer (GAD), with appealing results from several points of view [54,55,56]. 

In the present study, GAD was prepared according to the above cited procedure [54,55,56], and, in addition, novel GA dendrimer NPs were achieved by physical complexation of GA with the architecture of dendrimer **4**. 

A drug delivery system of GA (GALD, **7**) was realized, in which, differently from GAD, GA units are not chemically linked on the surface, but are located inside the dendrimer matrix cavities or absorbed onto its surface, where they are withheld mainly by hydrogen bond interactions. 

Concerning GAD, both the structure of dendrimer scaffold and of GA have been chemically modified by the mutual covalent bond. Differently, in GALD the structure of GA and dendrimer **4** were not modified during the synthetic pathway.

#### Chemistry 

Dendrimer **4**, herein adopted as nanocarrier to complex GA, and the GA-enriched dendrimer GAD (**6**) were prepared as previously described [51,52,53,54,55,56].

Concerning the novel GA delivery system (GALD, **7**), it has been achieved by dissolving dendrimer **4** in MeOH and by subjecting it to vigorously stir in the presence of a strong excess of GA (42.8 equiv.) for 48 h at room temperature in the dark (Scheme 1). 

Since, as reported in the literature [66] and Merck Index 2001, GA can be isolated by recrystallization from methanol, it was thought to purify the crude product, by removing the not entrapped free GA, through its precipitation from the final clear methanol solution. Briefly, the solution was concentrated until it became cloudy and a white solid started to separate. The precipitation was completed by placing the fine suspension at 4 °C overnight, and the obtained solid was recovered from the GALD solution by performing a centrifugation at 3500 rpm for 15 min.

GALD **7**, having the intuitive structure shown in Scheme 1, was achieved as off-white glassy solid (188.2 mg) and was stored on P_2_O_5_ in a dryer at r.t. 

In regard to Scheme 1 and Figure 3b, the 3D structure of the dendrimer **4** was gained, using the 3D Chem Draw software (Chem 3D Pro 7.0) and the mode of representing the complex host-molecule/dendrimer herein used, was accepted as valid in a recent publication, where dendrimer **4** was employed to complex ETO [57]. 

### 3.2. Physicochemical Characterization of Dendrimer **4** and GA-Enriched Dendrimers **6** and **7**

The physicochemical characterization of dendrimer **4**, not including the DLS analysis, and the complete characterization of GAD **6** have been previously reported [54], and the salient data re provided in Appendix A and Appendix A. Additional experiments to determine particle size, Z-potential, and the respective PDI value of dendrimer **4**, lacking in the previous study [54], were performed and the results are reported in Section 3.2.9.

GALD **7** was instead qualitatively characterized, by performing the colorimetric FeCl_3_ test for detection of phenol’s presence and by Fourier Transform Infrared (FTIR) spectroscopy. 

Data obtained from FTIR spectroscopy were also handled by performing Principal Component Analysis (PCA). Finally, Nuclear Magnetic Resonance (NMR) and UV–Vis analysis were carried out. UV–Vis spectrophotometric analysis associated to the Folin–Ciocalteu method was selected to evaluate the GA concentration in GALD **7** and to determine the drug loading (DL%) and the entrapment efficiency (EE%) of GALD. 

In addition, UV–Vis analysis was used to investigate the GA release profile while GALD’s particles were examined by Dynamic Light Scattering (DLS) analysis. 

#### 3.2.1. Colorimetric FeCl_3_ Essay

To confirm the success of the complexation reaction, the FeCl_3_ test for the detection of phenols was performed by comparing the coloration obtained on an ethanol solution of GALD (Appendix A, Appendix A) with that of a GA ethanol solution. 

As shown in Appendix A (Appendix A), the strong dark blue color of the GALD solution was completely analogous to that found in the GA solution. 

#### 3.2.2. FTIR Characterization

An exemplificative copy of the FTIR spectrum of GALD is shown in Appendix A (Appendix A). 

The FTIR spectra of dendrimer **4** and GA were acquired in the same conditions, for allowing the comparison between the spectra of the original components (**4** and GA) and that one of the complex (GALD). Copies of these spectra are available in Appendix A (Appendix A) and Appendix A (Appendix A). FTIR analysis confirmed the success of the reaction of complexation of GA with dendrimer **4**. 

Concerning the specific regions of absorption of GA in the range 4000–400 cm^−1^, Vijayalakshmi and Ravindhran [67] identified peaks at 1022, 1234, 1448, 1622, 1714, 3043, 3280, and 3365 cm^−1^. 

In another study, characteristic peaks at 3492, 3370, and 3282 cm^−1^ (OH groups), at 2920 and 2850 cm^−1^ (aromatics C–H stretching), at 1701 cm^−1^ (carbonyl absorption in carboxylic acid), and a band at 1615 cm^−1^, corresponding to stretching-related carbon–carbon in alkenes, were reported [68]. 

In the FTIR spectrum of GA herein acquired (Appendix A, Appendix A), bands detected at 3496 and 3228 cm^−1^ (OH groups), at 3061 cm^−1^ (stretching of C–H in aromatics), at 1688 cm^−1^ (conjugated carboxyl absorption), and at 1610 cm^−1^ (C=C stretching) were considered as significant. 

As shown in Figure 4 and Appendix A (Appendix A), in the FTIR spectrum of GALD **7**, several bands belonging to GA and not observable in the spectrum of **4** are instead well visible, thus confirming the presence of GA in the cavities of hosting dendrimer **4**. 

The peaks lists, obtained from the spectra in Figure 4, are included below. The pivotal peaks, indicating the successful complexation, are underlined.

**GA**. FTIR (KBr, cm^−1^): 3496, 3383 (shoulder), 3279, 3061, 1688, 1610, 1543, 1425, 1386, 1318, 1220, 1098, 1025, 901, 866, 767, 733, 634, 558, and 485. 

**4**. FTIR (KBr, cm^−1^): 3436, 2936, 1737, 1631, 1472, 1236, 1128, 1045, and 657.

**GALD 7**. FTIR (KBr, cm^−1^): 3600–3200 (large band OH of **4**), 3497, 3383 (shoulder), 3286, 2932 (**4**), 1734 (C=OO of **4**), 1688, 1615, 1425, 1320, 1028, 867, 734, and 559.

#### 3.2.3. NMR Characterization 

The successful complexation of GA with dendrimer **4** was further validated by ^1^H NMR spectroscopy. Since GA is a tetra-substituted phenyl derivative, in the ^1^H NMR spectrum, it presents a single signal in the aromatic protons zone (7–8 ppm) belonging to the two equivalent protons in *ortho* position to the carboxyl group (Appendix A, Appendix A). Differently, dendrimer **4**, adopted to entrap GA, does not have signals in this region of the spectrum because it does not encompass aromatic rings in its structure (Appendix A). The ^1^H NMR spectrum of GALD **7** (Figure 5 and Appendix A) presents both signals in the region 0–5 ppm, corresponding to the signals belonging to dendrimer **4** and a singlet signal at 7.15 ppm, matching the signal belonging to GA.

In the case of characterization of GAD, the ^1^H NMR analysis and the ratio between integrals values of significant selected bands were useful also for obtaining a quantitative evaluation of the GA units covalently linked to the dendrimer surface, with considerable precision. In the case of the characterization of GALD, it was supposed that the physical complexation process, which can occur both by inside encapsulation and by surface absorption, might hide some GA units in the internal cavities of the carrier, making them less detectable in the NMR analysis. 

Consequently, to avoid quantification errors, a more reliable and commonly adopted UV–Vis method, which quantifies the GA released by GALD in methanol solution, was performed to determine the amount of GA loaded by dendrimer **4**.

#### 3.2.4. PCA on FTIR Spectral Data

Concerning the FTIR prediction of chemical composition of GA-enriched formulation **7**, more reliable information was achieved, by performing the PCA on data from FTIR spectra of dendrimer **4**, GA and GALD.

In PCA, multi-dimensional data are reduced to a small number of new variables—principal components (PCs)—which are orthogonal linear combinations of the original ones that efficiently represent data variability in low dimensions [69].

Briefly, PCA is able to evidence similarities or differences among the samples under study by clustering or separating them within a square of two components identified for being Information carried out by PCs is expressed in terms of percentage of explained variance. By definition, PC1 has the largest percent explained variance, followed by PC2, PC3, etc. [55]. 

In this regard, the data of nine spectra obtained by the FTIR analysis made in triplicate on dendrimer **4**, GALD **7**, and GA, were organized in a large matrix of variables, which were processed by PCA. Overall, 99.3% of explained variance was provided by PC1 and PC2 and the results are reported as bi-plot in Appendix A (Appendix A). The wanted information was unequivocally observable on PC2. 

In this regard, by observing the relative positions of dendrimer **4**, GALD **7** and GA **1** on PC2, it appears that **7** is located in the first left square as GA, while the empty dendrimer **4** is located in a different square. Consequently, GALD, in terms of both location in PCA results and chemical composition, outcomes much closer to GA rather than to dendrimer **4**, suggesting that in the chemical structure of GALD **7** there is an unequivocal contribution of GA. By extrapolating the vectors up to intercept the PC2 axis, the GA loading can be estimated around 69%, i.e., higher than the expected 50%. The so high DL%, mathematically not allowed by the amounts of material employed in the complexation reaction, was confirmed by UV determination and relative explanations are discussed in the relative section.

#### 3.2.5. UV–Vis Spectrophotometric Analysis of GALD

The presence of GA in GALD formulation was confirmed also by UV–Vis spectrophotometric analysis. The UV–Vis spectrum of free GA in methanol shows absorbance maxima at λ = 218 and 273 nm (Figure 6, yellow plot), while the UV–Vis spectrum of empty dendrimer **4** complexing GA, in methanol, shows no peck of absorbance in the λ range considered (not presented results).

The UV–Vis spectrum of GALD, acquired in the same solvent and at the same concentration, after dissolution and release of GA from the dendrimer scaffold, showed two absorption peaks, obviously less intense, at the same λ = 218 and 273 nm (Figure 6, violet plot). 

In a study concerning GA complexes with iron [70], it was reported that, after complex formation, a bathochromic shift of the absorbance maxima of GA was observed, probably provoked by GA structural changes and/or by partial oxidation of GA caused by iron action. 

Since in this case, after complex formation between GA and dendrimer **4**, no bathochromic shift in λ_max_ values was observed, it can be asserted that no structural change or oxidation processes have been occurred. 

#### 3.2.6. GA Content Determination

For the estimation of polyphenols such as GA, the most reported method is the well-known colorimetric Folin–Ciocalteu method [63] based on the UV determination of the colored compounds deriving from oxidation of phenols, by the aqueous solution of phosphomolybdate and phosphotungstate (Folin–Ciocalteu reagent). 

Although unfortunately it leads to the loss of the sample [71], Folin–Ciocalteu analysis presents several advantages in terms of low costs of consumables, simplicity, and reproducibility [67,72]. The results obtained performing this method find confirmation in a recent study where GA content in GALD was quantified by performing a novel FTIR-based method considered accurate and robust [71]. 

##### Standard GA Calibration Curve by Folin–Ciocalteu Method

The GA solutions were analyzed by measuring the absorbance at λ = 760 nm, since it represents the absorbance maximum for the product of the reaction [72].

Equation (1) of the regression achieved proved a high value of R square (coefficient of determination, R^2^ = 0.9887), which suggested a condition of linearity.

The linearity and sensitivity of the developed calibration line were further evaluated, by confirming the statistical significance of its slope.

The linear regression correlating the C_GA_s and the C_GAp_s available in Appendix A (Appendix A) shows a R value of 0.9943, suggesting high correlation between the reference GA concentrations and those predicted by the calibration model and therefore asserting the good ability in prediction of the developed model.

The performance of the calibration model was evaluated also in terms of coefficient of determination (R^2^); standard error of calibration (SEC), i.e., the residual standard deviation of regression; relative standard deviation (RSD); standard deviation of the mean (SD_m_); root mean square error of calibration (RMSEC); and relative error of calibration (REC%). SEC and RMSEC were computed according to the Appendix A, respectively (Appendix A), while REC was computed according to the Appendix A (Appendix A). RSD was obtained by dividing SEC for the mean of C_GA_s, while SD_m_ by dividing SEC for the square root of samples number (*n* = 5). Data concerning errors in the calibration, R and R^2^ are reported in Appendix A.

In general, acceptable calibration models should have low SEC, RMSEC, and REC%, as well as high R and R^2^ values. In addition, an error of 10% in absolute term between the reference concentrations and the predicted ones is commonly accepted as the limit of the significance of the difference [73].

In this regard, the error values in absolute terms in the calibration set ranged from 0.05% to 0.3%. SEC and RMSEC were far lower than the acceptable limits (2%) of accuracy commonly adopted for analytical methods in the pharmaceutical industry [74,75]. REC% was good (5.3%) and R and R^2^ values were very high.

##### Estimation of GA Concentration in GALD **7** (DL%)

DL% was found to be 74.1% w/w and therefore far higher than expected, as reported in Table 1.

The molar extinction coefficient of GA released by the complex and oxidized (ε_GAOxC_) at the obtained concentration of 23.6 µg/mL was not significantly different from the molar extinction coefficient of free GAOx (ε_GAOx_), thus establishing that no significant variation (<1%) in the GA extinction coefficient with its complexation occurred, thus confirming the feasibility of the analytical method. The computed MW and EE% of GALD are reported in Table 1.

Concerning EE%, a percentage value higher than 100 could appear rationally impossible and, therefore, apparently unacceptable, but an EE% higher than 100% has already been reported for the encapsulation of coumarins in poly(lactic-co-glycolic acid) (PLGA)-based polymers, due to the poor solubility of polymers in the adopted solvent [76].

In the present case, such result is explainable by recognizing in dendrimer **4** a very high capacity to create interactions with GA molecules, thus managing to load a very high number of GA moles per dendrimer mole. This high capacity might derive by the presence on **4** surface of multiple hydroxyl groups, forming a sort of molecular tweezers, which might also allow an efficient surface complexation in addition to encapsulation. On the other hand, a very high value of EE% was expectable on the base of the previously obtained value of DL% (74%) and molar load (125.5). In this regard, PCA analysis prediction also had asserted a DL% of 68%, i.e., higher than expected and higher than the DL value allowed by the amounts of GA and dendrimer **4** employed for the complexation reaction (50%). 

Briefly, it can be supposed that the loading power of each mole of dendrimer was so effective that, once all the available GA moles were loaded, moles of dendrimer carrier resulted not complexed. In the phase of purification of GALD, while thinking to recover not complexed GA, these exciding moles of dendrimer were separated by crystallization from methanol solution, in place of GA, no longer present in solution. Consequently, GALD composition encompasses more GA than **4**. This assumption was validated by acquiring the FTIR spectrum of the isolated solid (Figure 7).

As observable in Figure 7, the spectrum of isolated solid matches perfectly that of dendrimer **4**, thus confirming the identity of two compounds. Further validation was obtained by PCA analysis on data from FTIR spectra of dendrimer **4**, GA, GALD, and the non-complexed substance recovered as solid from MeOH (Appendix A, Appendix A). 

In virtue of this phenomenon, a nanocomposite made in weight of less dendrimer than that employed in the reaction and therefore with a DL and EE higher than that initially possible was achieved. 

The high EE% and DL% translated into a very high estimated MW, which, according to what is reported in [61], characteristically extends retention time in systemic circle and affects positively the bio-efficiency of delivering nanodevices. 

#### 3.2.7. In Vitro GA Release Profile from GALD 

The in vitro release profile of GA from GALD **7** NPs into phosphate buffer saline solution at 37 °C and pH 7.4, mimicking human blood conditions, evaluated also to predict the GALD in vivo stability and its half-life, in reported Figure 8 as Cumulative GA release percentage in function of time. No action to stabilize GALD **7** or to influence GA release, as well as the polyester construction of dendrimer degradation, was performed. 

Drug release from carriers is influenced by several factors including the composition of the final delivery system, the ratio drug loaded/polymer scaffold, the possible physical and/or chemical interactions among the components, and the methods selected for preparing the formulations. 

Concerning dendrimers, drug release depends mainly on the type of interactions between the transported drug and the dendrimer structure and on the biodegradability degree/rate of the dendrimer scaffold. Generally, physically entrapped drugs are released more quickly than covalently conjugated drugs [77]. 

As shown in Figure 8, concerning GALD **7**, an initial slight burst release of GA can be observed and only 42% of GA was released in 24 h. 

Even if few papers have put forth theories to explain the phenomena of burst release, several studies report the presence of an initial severe burst release [78], as has been described for methotrexate (MTX) entrapped in hydroxyl-terminated G5 PAMAM dendrimers, which was 60% released in 2 h [77]. 

In our study, the burst release resulted significantly softer, asserting a higher stability of GALD if compared to other reported cases [77,78,79,80], and this is a great improvement, in terms of minor systemic toxicity, over most micellar carriers, such as PEGylated PLGA-based NPs that generally have a high initial drug release [79,80]. 

Probably, stronger physical interactions, such as more stable hydrogen interactions between the carboxyl and hydroxyls functions of GA and the polyester-based architecture with the 64 peripheral hydroxyls of dendrimer, managed to slow down the diffusion rate of GA through the polymer and to prevent its massive initial release.

Subsequently, a very slow and sustained GA release was observed and the total GA (98%) was released after 96 h, thus stating an in vitro half-life of 48 h for GALD. In this regard, it must be considered that drug carriers, such as biodegradable polymers, including polyesters, polyamides, poly amino acids, and polysaccharides, can release drugs via hydrolytic degradation of bonds in their backbones [78]. 

It was reported that hydrolytic degradation of polyester-based dendrimers can also simply occur in water or buffer solution at pH = 7.4–7.5 and temperature of 37 °C, starting after 6 h and arriving to completion in a few days, but it is faster in alkaline solution and slower at low temperature [81].

Therefore, the secondary slow and protracted release phase observed for GALD can be explained hypothesizing a degradation-controlled release and a slow degradation rate of the polyester-based scaffold of the dendrimer carrier at pH = 7.4 and at 37 °C, as reported [81]. 

Moreover, as already reported for the drug release from an analogous fifth-generation polyester-based dendrimer [80], the typical neutral value of pH of blood can favor the minor degradation rate and the slow systemic release. 

On the contrary, even if in contrast to what was observed by Feliu and co-authors (2012), according to another study [82] and thinking of an in vivo application of GALD on NB cells, the acidic environment of cancer cells, due to their production of lactic acid, can determine a faster in situ degradation and release, favoring a targeted action. In addition, since the degradation rate of the polymer depends also on molecular weight, end groups, monomer composition, and crystallinity, it is rational to think that a high generation and high molecular weight dendrimer as **4** could be subjected to long degradation times, thus determining a slow drug diffusion and a slow and protracted drug release. 

On the other hand, the rather total GA release (98%) suggests a complete disintegration of the carrier into small monomer compounds as previously described [81], whcih can more readily be removed from the body, allowing to minimize long-term side effects.

Finally, the initial faster release of GA, followed by a secondary slow and protracted release phase support the above-postulated presence of a part of GA associated at the surface and another part included inside of dendrimer. 

#### 3.2.8. Kinetics Release of GA from GALD Nanocomposite

The kinetic release of GA from the GALD nanocomposite was fitted using a number of different kinetic models, such as zero-order, first-order, Higuchi, Hixoson–Crowel, and Korsmeyer–Peppas models [83] (Figure 9). 

The results in Figure 9 establish that the release of GA from GALD **7** fits the Higuchi kinetic Model (Figure 9e). 

The criterion for selecting the most suitable model and describing the kinetic of the GA release profile of GALD was based on the higher value of the coefficient of determination (R^2^). In this regard, being its R^2^ = 0.9918, it was concluded that the GA release pattern of GALD was, among others models, best fitted with Higuchi model and follows Higuchi drug release kinetics. The Higuchi equation plot (Figure 9e) shows the release of the drug from the GALD matrix as a function of the square roots of time (SQRT) and therefore is dependent on Fickian diffusion. 

#### 3.2.9. Particle Size, Z-Potential and PDI of Dendrimer **4** and GALD

The size of NPs plays a key role in determining the performances of a nanosized delivery system (NDS).

It affects NDS distribution, toxicity, targeting ability, and drug release profile [84].

The preferential size, for successful biomedical applications and for assuring an efficient cellular up-take, has to be less than 200 nm, with an optimal of 100–200 nm [85,86].

Particles of 200 nm or larger promote the activation of the lymphatic system and consequently are quickly removed from circulation [85]. 

Particles with a size of about 100 nm allow a more efficient and faster drug release [87] and could also pass through the Blood–Brain Barrier (BBB) and deliver sufficient amounts of drugs, avoiding immediate clearance [84]. 

Particle sizes of dendrimer **4** and GALD were determined by DLS analysis, expressed as Z-AVE size (nm) and reported in Table 2.

Dendrimer **4** particles showed an average size of 45 nm with a satisfying PDI of 0.2, which does not denote possible formation of aggregates, confirmed by a sufficiently high Z-potential negative, assuring stability for dendrimer **4** aqueous solutions. On the other hand, NPs bearing too high surface charges, either positive or negative, attract easier macrophages.

Consequently, a Z-potential around 20 mV, as in the present case, may be promising for avoiding opsonins adsorption and subsequent clearance from the body by phagocytic cells. In addition, a Z-potential negative, probably due to the preferential absorption of hydroxyl ions on the uncharged surface of the dendrimer, hampers the binding with plasma proteins, which in turn may impede or reduce cellular up-take [88].

GALD particles showed an average size of 350 nm (Appendix A, Appendix A), i.e., a dimension superior to the reported optimal value of 100 nm [84] and uncommon for dendrimers of similar generation [89], but in strong accordance to particle size of GAD (Appendix A). 

As previously reported for GAD [54], such phenomenon is attributable to the possibility for dendrimers to form dendrimer multi-molecular assemblies, known as megamers, which are typically characterized by similar dimensions [90]. The formation of stable megamers can be intentionally provoked by introducing cross-linking agents during the dendrimer synthesis. Otherwise, reversible megamerization can occur because of the spontaneous assemblies of dendrimer molecules into supramolecular assemblages. 

In the present case, such assembly process may have been reasonably favored by the several hydroxyl groups, of both GA units and the dendrimer scaffold, which can establish many hydrogen bonds between the unimolecular dendrimer molecules, thus giving rise to dendrimers aggregates. 

However, as observable in Appendix A (Appendix A), particles with smaller dimensions of 119 nm, which matches the ideal value of 100 nm, are detectable [84].

In this regard, GALD nanoformulation, thanks to the physical properties deriving from its particles size, could possess a unique biologic potential for biomedical applications. As reported in [91], drug delivering devices of such dimensions represent attractive systems for treatment of cancer and heart and lung, blood, inflammatory, and infectious diseases, including central nervous system disorders. As expected, the polydispersion index (PDI) value was rather high (0.7), thus confirming the presence of particles of different sizes (100–400 nm) in water solution. Differently, the value of the zeta potential (−29 mV) (Appendix A, Appendix A) was higher than expected and indicated a certain stability of GALD solutions.

Usually, high PDI values translates into very low Z-potential values and particles tending to agglomerate show Z < 5 mV that means physical instability in solution. On the contrary, a Z-potential near ±30 mV, either positive or negative, usually leads to low PDI values, absence of aggregates and good physical stability in solution [92,93]. 

In this regard, GALD particles, tending to agglomerate and having a high PDI should be endowed with a Z-potential value lower than that measured. This apparent anomaly is explainable with an incessant process of aggregation/disaggregation of the particles, thanks to which the critical concentration of large particles to determine precipitation is never achieved.

It is consequently conceivable that in vivo this harmonic balance will also allow avoiding a massive phagocytic response of macrophage cells. Finally, since high positive Z-potential values are correlated to high cytotoxicity and cells internalization, while negative values allow less damage for the physiological membranes [94], the negative Z-potential value of −29.2 mV ± SD suggests low systemic toxicity and low cytotoxicity on health cells. 

In addition, although NPs with positive Z-potential are commonly promptly absorbed on cells surface by electrostatic interactions, and are internalized more easily than ones with negative Z-potential, positive Z-potentials also favor the adsorption of negative albumin that, therefore, hampers a subsequent interaction with cell membrane and internalization [88].

In this regard, it was found that NPs with negative Z-potential, able to repel serum proteins, was favorable for the nanoparticles uptake in tumor cells.

#### 3.2.10. Evaluation of GALD Solubility

According to literature data, GA is well soluble at room temperature in DMF; sufficiently soluble in MeOH, ethanol, and acetone; and very poorly soluble in water (11.01–11.56 mg/mL [95], 15.3 mg/mL [96]) and ethyl acetate (13.09 mg/mL [96]).

To evaluate the influence of having resizing GA at nanodimensions on its poor solubility, the solubility of GALD was determined at room temperature, in both water and ethyl acetate, following a procedure previously described for polyester-based dendrimer drug formulations and accepted as valid [37,57]. 

Then, knowing the content of GA in GALD complex, it was possible to calculate the actual amount of GA that was dissolved in 1 mL of water and ethyl acetate, when 6.3 mg of GALD exactly weighted were dissolved.

The so-obtained solubility of GALD as a complex, and of GA loaded in GALD, expressed as mg/mL, is reported in Table 3 and compared to the solubility data of free GA reported in the literature [95,96]. 

GALD **7** gave clear and stable solutions in both water and ethyl acetate at concentrations of 126 and 31.5 mg/mL, respectively. By considering that GA DL% was 74.1%, the exact amount of GA that was possible to dissolve in water and ethyl acetate through the dissolution of GALD was 93.4 and 23.4 mg/mL, respectively. Interestingly, the solubility of GA in water, which is a biocompatible solvent for drug administration, was improved by 6.1–8.5 times.

### 3.3. Time-Course and Dose-Dependent Experiments on GA

Since it was the first time that GA was considered as alternative possible compound to treat human NB, time-course dose-dependent experiments were performed to evaluate its effects on NB cells viability and intracellular ROS induction.

The activity of free GA, both on ROS production and on NB cells viability, in function of GA concentrations (10–150 µM) and time of cells exposure (48 and 72 h) were assayed both on HTLA-230 and on HTLA-ER NB cells [12] and the results are reported in Figure 10 and Figure 11.

As reported in Figure 10, a significant improvement of ROS production was observable at concentrations of 100 and 150 µM for HTLA-230 and HTLA-ER cells, respectively, after 48 h (Figure 10a), and at concentrations of 75 and 100 µM for HTLA-230 and HTLA-ER, respectively, after 72 h (Figure 10b), thus confirming a time dependent pro-oxidant action.

Accordingly, a significant reduction in cells viability was observed at concentrations of 100 and 150 µM for HTLA-230 and HTLA-ER cells, respectively, after 48 h (Figure 11a), and of 75 and 100 µM for HTLA-230 and HTLA-ER, respectively, after 72 h (Figure 11b). 

Both the GA-induced ROS production and cytotoxicity were more marked in the cells sensitive to ETO than in the resistant ones, which have developed enhanced antioxidant defenses [12].

These findings confirm that also GA cytotoxic activity depends on length of cell exposure and suggest a cause–effect relationship between ROS production and cytotoxicity. 

In fact, this correlation, extensively reported for phytochemicals such as GA [97], was further confirmed by the exact overlap of the active concentrations of GA able to cause both a significant increase in ROS and a significant decrease in cell viability at 48 and 72 h.

In addition, as reported for many other kinds of tumors [98], these findings confirmed that, also on NB cells, GA exerts a ROS-mediated cytotoxic anticancer activity at high doses, while it loses its pro-oxidant properties at low doses [97]. 

At doses lower than 75 µM, GA did not influence significantly ROS production or viability of both NB cells populations, further confirming that ROS production and the cytotoxic action are correlated by a cause–effect relationship.

Although the concentrations of GA active on NB cells, including resistant ones, appear very high, they are in accordance or lower than those reported for GA effectiveness on other tumors [98]. In addition, phenolic compounds are established to be safe and not toxic even at higher doses [99]. In this regard, experimental sub-chronic toxicity assessment of GA in rats showed that animals fed with a diet containing increasing amounts of GA for 13 weeks do not exhibited toxic symptoms even at 119 (male) and 128 (female) mg/kg/day [100]. Oral administration of a dose up to 5000 mg/kg in mice was also found to be safe [101].

Despite these findings, and considering that the empty dendrimer **4** has a ROS-mediated cytotoxic activity on NB cells sensitive to ETO at the lower dose of 0.169 µM [57], instead of considering, as starting point, to plan the experiments on GAD and GALD, the concentrations of active of GA, i.e., that of dendrimer **4**, were considered. 

Thus, it was possible to assess if, by reformulation in NPs, GA could be active at lower doses and if dendrimer **4** is active also on HTLA-ER cells.

### 3.4. Evaluation of Cytotoxic Action of GAD, GALD, GA and Empty Dendrimer **4** on Human Neuroblastoma Cells

HTLA-230 and HTLA-ER NB cells [12] were treated with dendrimer **4** at the reported active concentration of 0.169 μM [57], GALD at a concentration capable of providing 0.169 μM dendrimer **4**, and GA at the concentration (21.20 μM) provided by the amount of GALD used. Then, to get the same concentration of GA provided by GALD, since GAD contains 64 GA units, its concentration was computed by dividing by 64 the concentration of 21.20 μM, which was 0.3313 μM. The cells were exposed to dendrimer **4**, GALD, GAD, and GA for 48 and 72 h and the effect of all treatments on NB cell viability was investigated. As reported in Figure 12a, 48 h of exposure to dendrimer **4** reduced HTLA-230 and HTLA-ER cell viability by 40% and 20%, respectively. Interestingly, comparing the effects induced by the 72-h treatment, dendrimer **4** further reduced by 5% the cell viability of both cell populations (Figure 12b). As expected, free GA did not affect cell viability after 48 and 72 h, due to the low dose employed, in both cells populations. 

Unexpectedly, at the used concentrations providing the amount of dendrimer **4** active when administered alone, the exposure to GAD or GALD did not affect cell viability after 48 and 72 h in both cell populations, highlighting instead that the presence of GA in the GAD and GALD formulations totally prevented the cytotoxic action of dendrimer **4** (Figure 12a,b). 

These results demonstrate that dendrimer **4**, in addition to inducing death in NB cells sensitive to ETO [57], is able to exert cytotoxic effects also in chemoresistant NB cells population.

In this regard, dendrimer **4** represents a nanodevice suggestable either as a promising novel therapeutic molecule, able to induce death, in both sensitive and resistant NB cells at low dose, and/or as a carrier for chemotherapeutic drugs, for realizing synergistic therapies and reducing drugs dosage. 

As a confirmation for the feasibility of this strategy for treating NB, recently it has been demonstrated that the encapsulation of ETO into the dendrimer **4** enhances ETO activity in a time-dependent way and facilitates its protracted release [57]. 

The results herein obtained in GA-treated cells at concentrations sub-active of GA confirm those obtained in the time-course and dose-dependent experiments, which show that, as reported previously [23,24], GA cytotoxic action is dose dependent and is performed only at high dosage, similar to many natural antioxidant–pro-oxidant compounds [97]. 

In recent years, to reduce active dosage of such compounds, the application of nanotechnology has been suggested [102,103], but, in our context, this approach was not successful. 

Although GA ROS-mediated cytotoxicity at low dose was not improved, by reformulating GA in NPs interesting findings and an unexpected goal were achieved.

Through the reformulation of low concentrations of GA in NPs, by using bioactive concentrations of dendrimer **4** and two different synthetic strategies, two nanosized forms of GA were achieved that, surprisingly, proved to exert such remarkable antioxidant effects as to be able to completely abolish the pro-oxidant and cytotoxic activities of the dendrimer.

### 3.5. The Presence of GA Nullifies the Pro-Oxidant Action of Dendrimer **4** in NB Cells Exposed to GAD and GALD

As shown in Figure 13, dendrimer **4** increased ROS production in both cell populations, although this effect was more evident in HTLA-230 than in HTLA-ER cells. In particular, compared to control cells, in HTLA-230, ROS levels were increased by 46% and 63% after 48 (Figure 13a) and 72 h (Figure 13b), respectively, while, in HTLA-ER cells, the same treatments stimulated ROS generation by 30% and 43% after 48 (Figure 13a) and 72 h (Figure 13b), respectively. On the contrary, in both cell populations, the exposure to free GA did not change ROS production at 48 or 72 h (Figure 13a,b), as in the case of cells viability, due to the low dose employed. Although it has been previously demonstrated that GAD possesses a remarkably antioxidant action higher than GA [54,55,56], in these cell populations, only a slight potentiation of antioxidant power of GA was observed for both GAD and GALD (Figure 13a,b). The strong antioxidant activity of GA, when reformulated in NPs, was distinctly highlighted by the finding that it was able to nullify ROS overproduction induced by the dendrimer **4** in both cell populations, at low dose and either at 48 and 72 h (Figure 13a,b).

Returning to dendrimer **4**, thanks to its pro-oxidant activity, it was able to create a condition of OS responsible for triggering NB cell death also in chemoresistant populations. 

Since NB cells, and in particular chemoresistant ones, have been demonstrated to activate antioxidant responses [12,13,104], the induction of OS could be efficiently employed to sensitize cancer cells to pro-oxidant cytotoxic action of chemotherapeutic drugs. In this regard, we have recently demonstrated that the combination of ETO with dendrimer **4** by increasing the pro-oxidant action of ETO is able to sensitize not resistant NB cells to the drug [57]. 

This strategy could become useful in vivo to enhance sensitivity of tumors to antineoplastic agents by lowering drug’s cytotoxic doses and therefore their systemic toxicity. 

## 4. Conclusions

The interesting results from the present study suggest that novel promising strategies can be developed for treating cancer, by applying nanotechnology and natural compounds. In regard to NB, to date not yet faced with GA, GA was able to exert a ROS-mediated cytotoxic action, both on sensitive and on chemoresistant NB cells, only at high concentrations (75–150 µM), and it was considered safe. Interestingly, the nanoengineered dendrimer **4**, previously found to possess a ROS-mediated cytotoxic activity on sensitive NB cells, in this study showed to be able to chemosensitize also NB cells resistant to ETO at a concentration almost 100-fold lower than that of GA. 

In this regard, dendrimer **4** nanodevice is advisable, either as a promising novel therapeutic drug effective also on cells resilient to available drugs or as an advantageous and synergistic carrier of chemotherapeutic drugs, thus allowing reducing their dosage, systemic toxicity, and side effects. Moreover, dendrimer **4** represents for researchers a useful template molecule for the development of a variety of analogous dendrimer devices endowed with cytotoxic activity against NB cells. 

Differently, the GA-enriched dendrimers GAD and GALD, administered at concentrations providing bioactive amounts of dendrimer **4** and concentrations of GA not able to induce ROS-mediated cells death, proved to be ineffective against NB.

Synthetizing and assaying GAD and GALD was not fruitless, because it allowed observing that the presence of GA, in GAD and GALD formulations, slowed down (HTLA-ER cells) or nullified (HTLA230) the pro-oxidant activity showed by dendrimer **4** when administered alone. 

In NB cells, this GA behavior hindered the ROS-mediated anti-cancer effect of dendrimer **4** against NB cells, but suggests a considerable ability of GA in counteracting ROS overproduction and OS, induced by **4**, also at low doses, when resized in form of NPs. 

These findings could have important clinical repercussions for two reasons. On the one hand, the results advise scientists that it is crucial to be careful to administer nanoformulated polyphenol-enriched food supplements to cancer patients treated with pro-oxidant chemotherapeutic drugs, because polyphenol NPs could invalidate their effectiveness, as GA versus dendrimer **4**.

On the other hand, the above findings evidence that, when reformulated in NPs, GA takes on the ability to inactivate compounds able to induce ROS and OS, which are the major causes of DNA damage and cancer initiation in healthy cells. 

It is clear that each of the compounds investigated in this work (GA, dendrimer **4**, GAD **6**, and GALD **7**) deserves further and more in deep investigations to clarify its mechanism of action at the molecular level. In this context, experiments on dendrimer **4**, in order to investigate the metabolic pathway responsible for ROS overproduction and NB cell death, are already underway. In addition, another polyester-based scaffold, analogous of **4** but slightly modified in the structure, is also under study, in order to find how structural modifications could influence cytotoxic activity. 

In addition, thinking of a future clinical application of dendrimers, it is the duty of the authors to remember the general concern regarding the potential dangerous effects of nanomaterials on human health. In addition to existing in various sizes, shapes, and chemical compositions, NPs occur also in different degrees of agglomeration. Intracellular distribution is affected by agglomeration and agglomeration may represent a risk factor because it enables NPs to accumulate within the tissue, thus reducing clearance efficiency and improving toxicity. In this regard, knowledge regarding agglomeration-based accumulation and its relevance is still limited. The currently available in vitro literature is controversial, and findings regarding the effect of agglomeration are poor, thus it is strongly recommended to include these aspects in future investigations.

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
