# Peer review of "Dendrimer Nanodevices and Gallic Acid as Novel Strategies to Fight Chemoresistance in Neuroblastoma Cells"

_nanomaterials, 2020, doi:10.3390/nano10061243_

Round 1

Reviewer 1 Report

In this publication authors utilized earlier developed dendrimers, encapsulated, and functionalized with gallic acid. This publication provides some useful information’s for the treatment of chemo-resistant neuroblastoma. The authors have performed extensive studies to support the findings. However, few improvements are required. I would also advise the authors to use an advanced method such as HPLC for the quantification of encapsulated molecules because UV-Vis measurements are not accurate.

Some of the corrections that need to be done:

Line 874-875: Revise it as it is not understandable.

Line 596: % EE was observed to be 148.4% which is practically impossible. However, an explanation of this behavior should be discussed clearly with proper references.

Figure 12: In some bar graphs cell viability seems to be more than 100 which is not possible. Check again and correct it.

Figure 10-13: Y-Axis title is absent in these figures. Should be modified.

Figure S10: This figure is reported in the earlier publication. Reference should be provided.

Figure S11: This figure is also reported in the earlier publication. Reference should be provided

Figure S9: Axis titles are missing.

Author Response

REVIEWER #1

In this publication authors utilized earlier developed dendrimers, encapsulated, and functionalized with gallic acid. This publication provides some useful information’s for the treatment of chemo-resistant neuroblastoma. The authors have performed extensive studies to support the findings. However, few improvements are required. I would also advise the authors to use an advanced method such as HPLC for the quantification of encapsulated molecules because UV-Vis measurements are not accurate.

Concerning the request of using HPLC in place of UV-Vis technique for quantifying GA in GALD, the authors make kindly note that, even if HPLC is a well performant technique widely used, they have adopted UV-Vis technique associated to Folin Ciocalteau method, because it is the most adopted method to determine the content of GA in different polymer or very complex matrices, as foods or vegetable extracts.

In this regard, please see Refs. 63, 64, 67, 71 and 72 cited in the manuscripts.

The method is frequently preferred to HPLC, because more simple and low cost. Moreover, UV-Vis method, in addition to be simple and low-cost, it is considered also reliable and able to provide reproducible results. Please see Refs. 67 and 72 cited in the manuscripts and:

Felipe Hugo Alencar Fernandes & Hérida Regina Nunes Salgado (2015). Gallic Acid: Review of the Methods of Determination and Quantification. Critical Reviews in Analytical Chemistry. DOI: 10.1080/10408347.2015.1095064.

Moreover, it was published recently a study concerning the assessment of a FT-IR method to quantify GA in different polymer matrices (Alfei et al. 2020, Ref. 71 cited in the manuscript), in which GALD was used among others, as sample test to validate the developed method, which resulted robust, precise and accurate. In this regard, the findings obtained from the quantification of GA in GALD with this novel method were in accordance with those obtained by using UV-Vis method, thus confirming the accuracy of the method.

Anyway, to clarify this question, some additional sentences have been included in the revised manuscript. Please see lines 506-508 of the revised manuscript.

Considering all these argumentations, the authors ask the Reviewer to settle for the UV determination.

Some of the corrections that need to be done:

Line 874-875: Revise it as it is not understandable.

As requested, the part of the manuscript suggested by the Reviewer has been revised, in order to make it more understandable. Please see lines 879-888 of the revised manuscript.

Line 596: % EE was observed to be 148.4% which is practically impossible. However, an explanation of this behavior should be discussed clearly with proper references.

An explanation by the authors was already present in the unrevised version of the manuscript. Please see lines 613-637 of the revised manuscript.

Anyway, in order to satisfy the Reviewer’s request, an additional explanation and the related reference (Ref. 76) has been included in the revised manuscript. Please see lines 609-612.

Figure 12: In some bar graphs cell viability seems to be more than 100 which is not possible. Check again and correct it.

The authors thank the Reviewer for noticing this apparent anomaly and for requesting a check. Consequently, the absorbance data provided by the microplate reader and the resulting cell viability (%) have been carefully checked. The results were confirmed and are precisely those shown in the graphs. The authors cannot correct what is experimental data.

Anyway, the authors provide the following explanation.

In some bar graphs cell viability results to be more than 100 since the value of cell survival in untreated cells (Ctr) is arbitrarily reported to 100 and therefore when a treatment is able to stimulate cell proliferation or reduce cell death, an increased percentage of cell viability is registered.

Figure 10-13: Y-Axis title is absent in these figures. Should be modified.

As requested, Figures 10-13 have been modified by inserting the Y-axis title.

Figure S10: This figure is reported in the earlier publication. Reference should be provided.

The reference of the work in which the Figure is reported was already present in the references list of SM, but the number was missing in the text near the Figure. As requested, the reference number [3] has been inserted in the Figure caption. The same operation was made by the authors also in the caption of Figure S12 [5]. These changes have not been highlighted, because Nanomaterials requires only the pdf. definitive version of SM, during submission. 

Figure S11: This figure is also reported in the earlier publication. Reference should be provided

The reference of the work in which the Figure is reported was already present in the references list of SM, but the number was missing in the text near the Figure. As requested, the reference number [4] has been inserted in the Figure caption. This change has not been highlighted, because Nanomaterials requires only the pdf. definitive version of SM, during submission. 

Figure S9: Axis titles are missing.

Authors are sorry to point this out, but the title of the x-axes in Figure 9, (i.e. Time (h), Square root of time (SQRT) or Ln e time) was already present in the unrevised manuscript. Please, kindly look better at Figure 9.

Reviewer 2 Report

1)           Based on the results, the authors should rewrite the manuscript to refocus it on the action of dendrimer nanodevices instead of highlighting the protective role of gallic acid.  The dendrimer mechanism of action should be investigated.

2)            A second method to measure cellular viability should be used in addition to the one reported by the authors. Many artifacts have been reported when methods based on tetrazolium salts have been used to measure cellular ability.  I suggest using Hoechst 33342 or Trypan blue as a second method to confirm the viability data.

3)            2′-7′-dichlorofluorescein-diacetate (DCFH-DA) has been used to measure oxidative stress but this dye is not specific for hydrogen peroxide since there are several reports indicating its reactivity with superoxide anion. Please confirm the hydrogen peroxide production by the use of a more specific dye: ie. The cell-permeable named Peroxy Orange 1.

4)            The authors should explain how the gallic acid is released in phosphate by incubation at 37ªc in the absence of an esterase that could break the ester bonds and release GA from the dendrimer. The authors should perform these assays in the buffers in the absence of metals that could degrade the dendrimer i.e. buffer washed with CHELEX or buffer supplemented with EDTA.

5)            Section “3.2.6. GA content determination” should be moved to the Materials and Methods as well as “3.2.6.1. Standard GA calibration curve by Folin Ciocalteu method” and 3.2.6.2. Estimation of GA concentration in GALD 7 (DL%) but results

6)            Please from all the shown models in section 3.2.8, select one to fit the data. In the case of figure 9 D, please fit the curve to the data.

7)            Section 3.3. Preliminary experiments should be eliminated from publication. Only the final data should be reported. The authors should complete the data or eliminate this section.

8)            The tile should be changed to focus it on the action of the dendrimer, highlighting the protective role of gallic acid

Minor changes:

Table 1 should be moved to the Supplementary Material

Figure 10, 11,12, and 13 are lacking Y-axis labeling.

Abstract, Include the meaning of these abbreviations in the abstract NPs, OS

The authors should indicate the composition of the dendrimer center (named as G5 in the picture)

Author Response

REVIEWER #2

1)           Based on the results, the authors should rewrite the manuscript to refocus it on the action of dendrimer nanodevices instead of highlighting the protective role of gallic acid. The dendrimer mechanism of action should be investigated.

The authors accept the opinion of the Reviewer, but they think that the properties of all the compounds tested in the present work have been highlighted equally.

Gallic acid (GA) has been highlighted for its pro-oxidant and ROS-mediated cytotoxic activity at high dose, anyway not toxic for health cells as documented (please see Refs. 98, 99 and 100, cited in the manuscript). The dendrimer nanodevice for its ROS-mediated cytotoxic action at low doses and the GA-enriched dendrimer nanoformulations, for having allowed GA to exert antioxidant effects so remarkable, to totally nullify the pro-oxidant effect of the dendrimer carrier, even at very low doses.

Anyway, in order to satisfy the Reviewer, changes and additions, along the entire manuscript (parts written in red ink in abstract, discussion and conclusions), have been made to emphasize nanodevices more.

Concerning the second request of the Reviewer, the author explain that, to investigate in deep and at molecular level, the mechanism of action of the dendrimer, was not in the scope of the present manuscript, which is (in their opinion) already very articulate.

The aims of the present work were the following.

1) to synthetize and characterize a novel GA-enriched formulation (GALD), structurally different from a previous synthetized one (GAD) in order to have two types of nanodevices made of a dendrimer bioactive fraction and GA differently connected, to investigate and compare.

2) to evaluate the antitumor activity of a small natural molecule (GA) known for its cytotoxic properties, but never tested on human NB cell line resistant to etoposide, a clinical-used drug. Of the synthetic dendrimer nanodevice (previously found active against NB sensitive cells) against NB resistant cells, and of the two dendrimer nanoformulations enriched with GA.

3) to evaluated if, analogously to etoposide, a correlation between the cytotoxic activity observed and the production of oxidative stress (ROS) exists, in order to be able to assert a ROS-mediated mechanism of action.

All these points have been addressed in the manuscript.

In addition, as it is also said in the conclusions, more in-depth investigations, such as those requested by the auditor, are already underway, both on the dendrimer herein reported and on analogues with different numbers of hydroxyls on the surface, trusting that the observed activity may depend on these numerous functions capable of produce radical species. Please see lines 970-976 in the Conclusions (highlighted in yellow).

2)            A second method to measure cellular viability should be used in addition to the one reported by the authors. Many artifacts have been reported when methods based on tetrazolium salts have been used to measure cellular ability.  I suggest using Hoechst 33342 or Trypan blue as a second method to confirm the viability data.

While on one hand the use of tetrazolium-based assays may give artifacts, on the other the high number of studies reporting the same assay (Duan et al., Cancer Letters 2010; Soman et al., J Immunol Methods 2009; Willems et al., J Vir Met 2011; Shi et al., BMC Cancer 2016; Gautam et al., Mol Cancer 2016; Gao et al., Cell Oncol 2018; Akter et al., Antioxidants 2019; Zhang et al., Cancer Cell Int 2019), lead us to believe, that the method herein used can be considered valid for evaluating cell viability.

Furthermore, although the authors would be available to carry out further experiments to confirm the results reported in the manuscript, due to the ongoing health emergency, the access to the laboratories is limited and they cannot carry out further assessments in the limited time of five day at disposition for completing their revision.

Concerning all these argumentations, the authors kindly ask the Reviewer to settle for the results obtained with the tetrazolium-based assays.

3)            2′-7′-dichlorofluorescein-diacetate (DCFH-DA) has been used to measure oxidative stress but this dye is not specific for hydrogen peroxide since there are several reports indicating its reactivity with superoxide anion. Please confirm the hydrogen peroxide production by the use of a more specific dye: ie. The cell-permeable named Peroxy Orange 1.

The authors agree with the Reviewer that the use of DCFH-DA for H2O2 determination is not a suitable probe, due to other several radical species contributing to the DCF accumulation. However, Cossarizza A. et al. (Nature Protocols 2009) reported that "DCFH-DA has a good specificity for H2O2, and it has been shown that the fluorescence of the product DCF appears to be mediated mainly by H2O2 ". In addition, the method that the authors use in this manuscript involves the lysis of cells in a solution of 90% DMSO and 10% PBS as suggested by Wang et al. (Free Radic. Res., 2009). In this study, Wang and colleagues report as follows: "Even though oxidation of intracellular DCFH to DCF may be affected by many factors and may not always be a measure of ROS formation, it nonetheless remains a useful and widely used probe of ROS production in cells. Cell lysis with 90% DMSO/10% PBS and the resulting homogenization and stability of the DCF fluorescence signal clearly results in a more precise and convenient measure of DCFH oxidation to DCF than other methods commonly used". Moreover, although the reported method is widely used in order to evaluate ROS and specifically H2O2 production (van de Wier et al., FEBS J 2013; Zhang et al., Cancer letters 2013; Martins et al., Autophagy 2018), we have changed the title of paragraph 2.12. "Detection of hydrogen peroxide (H2O2) production" in "2.12. Detection of reactive oxygen species (ROS) production" (line 337) and similarly in the description of the method, H2O2 was replaced by ROS (line 338).

4)            The authors should explain how the gallic acid is released in phosphate by incubation at 37ªc in the absence of an esterase that could break the ester bonds and release GA from the dendrimer. The authors should perform these assays in the buffers in the absence of metals that could degrade the dendrimer i.e. buffer washed with CHELEX or buffer supplemented with EDTA.

The authors thank the Reviewer for its interest in the architecture of dendrimer carrier, which, being a polyester-based molecules might require an esterase enzyme to completely hydrolyze and degrade its structure.

Anyway, they kindly make present that no esterase is necessary for detaching GA from dendrimer, since it is not linked to dendrimer through an ester-type bond, but only physically entrapped or absorbed on dendrimer surface.

GA is retained through hydrogen bonds made possible by the reciprocal structures of GA and of the dendrimer (please see lines 165-167, 355-357, 359-360, 654-655 and 665-668, highlighted in yellow).

As a consequence, the release of GA (described in the text as “slight burst release”, line 656, highlighted in yellow) can initially occur also in absence of an esterase, as already described for PAMAMs-type dendrimers PAMAM, known for not being attackable by esterase and for not being biodegradable (please see lines 654-655 and 658-661 and ref. 77, mentioned in the main text) and also for biodegradable polyester-based dendrimers (see lines 670-672, ref. 78, and lines 675-677, ref. 80, highlighted in yellow).

In this phase, the release of GA from the dendrimer occurs through rupture by the solvent (buffer at neutral pH at a temperature of 37 ° C to mimic the physiological conditions) of the dendrimer/GA interactions and swelling of the dendrimer, which creates new interactions with the solvent itself.

Subsequently, concerning biodegradable dendrimer and polymer supports, they can release transported drugs, through their slow hydrolytic degradation (as it has been reported in the text, lines 669-677), which occurs even in the absence of esterase enzymes and in simple water (see ref. 78 and 80 cited in the manuscript). In this regard, we quote a phrase reported in the work of Huang and Brazel, 2001 (Ref. 78 cited in the manuscript):

“In the work of Ahmed et al. [27], drug release from biodegradable PLGA microparticles prepared by a w/o/w emulsion method was characterized by a high initial burst release of about 60%, which was attributed to diffusion of the drug through pre-existing pores and channels in the microparticles formed during the solvent evaporation process. The remainder of the drug was released slowly as the PLGA eroded in water.”

In addition, it should be noted that, as suggested by the Reviewer, the action of an esterase is instead necessary for detaching GA from formulations in which it is covalently linked, as GAD. Indeed, in GAD, GA is linked to the surface hydroxyls of the dendrimer with ester bonds. In this regard, please note the work (Alfei et al. 2020) mentioned in the manuscript (Ref. 54), where the GA release tests were reported in the manner suggested by the Reviewer, i.e. in the presence of a pig esterase.

Concerning the Reviewer suggestion of using buffer supplemented with EDTA, the authors regret having to point this out, but the release experiments have already been done in PBS with the addition of EDTA, as he suggests (please see lines 294-298, highlighted in yellow).

5)            Section “3.2.6. GA content determination” should be moved to the Materials and Methods as well as “3.2.6.1. Standard GA calibration curve by Folin Ciocalteu method” and 3.2.6.2. Estimation of GA concentration in GALD 7 (DL%) but results

The authors have some difficult to understand the Reviewer request because his sentence it was not finished.

Anyway, the authors explain as follows.

The short paragraph 3.2.6. does not report experimental data but a brief discussion concerning the method, so that the author have considered a rational choice including it in Results and discussion section.

Concerning 3.2.6.1. and 3.2.6.2. sections, the authors agree with the Reviewer that some experimental details already reported in Material and Methods section have been repeated again and they have been removed (lines 511-516 and 566-571), but the parts containing and discussing the results and the statistical significance of the calibration method have been left in the results and discussion section.

6)            Please from all the shown models in section 3.2.8, select one to fit the data. In the case of figure 9 D, please fit the curve to the data.

The authors makes note to the Reviewer, that a model to fit the data of the release curve, has been already chosen and it is the model in Figure 9c. In addition in the text was already included an exhaustive explanation of why model in Figure 9c fits the data (lines 700-707, highlighted in yellow).

Anyway, being in agreement with the Reviewer that the term “fit” is more appropriate than the terms used (“tailored” or “the release followed”), they have been adequately replaced (see lines 692, 700 and 705).

7)            Section 3.3. Preliminary experiments should be eliminated from publication. Only the final data should be reported. The authors should complete the data or eliminate this section.

The author agree with the Reviewer concerning the use inappropriate of the term “preliminary” because they make appear incomplete data. Differently, data are already complete and the relative discussion detailed.

“Preliminary” has been removed along all manuscript (please see lines 168, 800, 802 and 890).

8)            The tile should be changed to focus it on the action of the dendrimer, highlighting the protective role of gallic acid

As requested the title has been changed in “Dendrimer nanodevices and gallic acid as novel strategies to fight chemoresistance in neuroblastoma cells and protect health ones”.

Minor changes:

Table 1 should be moved to the Supplementary Material

As requested Table 1 has been moved in SM where it appears as Table S2 (Section S7). Consequently, the Table numbers have been reduced by one in the main text and improved by one in SM.

Figure 10, 11,12, and 13 are lacking Y-axis labeling.

The authors apologize to the reviewer for their forgetfulness. The Y-axis labeling has been inserted in Figures 10, 11, 12 and 13, as requested.

Abstract, Include the meaning of these abbreviations in the abstract NPs, OS

As requested, the abstract has been modified, but instead of including the meaning of the abbreviations, abbreviations have been replaced by the complete definitions, in order to respect the number of words wanted by Nanomaterials.

The authors should indicate the composition of the dendrimer center (named as G5 in the picture)

Regarding the latter request of the Reviewer, the authors admit that they would have liked to insert in the image (Figure 2) the structure of the central part of dendrimer 4, but for reasons of space, the sphere was used with the number of the generation indicated (G5). This polyester structure is however observable in a stylized form in the simplified structure of GAD (Figure 3), which is made of the same scaffold and in the structure of the dendron called D5COOH in Figure S1 in SM, which has been used to esterified 1,3-propandiol to provide dendrimer 4.

In addition, this way of representing the structure of dendrimer 4 has been already used in a previous work (Ref. 57) and accepted as valid. 

Anyway, sentences explaining this question have been included in the caption of Figure 2. Please see lines 138-141.

Round 2

Reviewer 2 Report

Although the authors answered all my questions I remain caution about potential pitfalls in this study. 

1) To change the manuscript orientation is a suggestion based on the author's results but I agree with the author's response, the authors are the ones to decide how to orient the manuscript in one or another way. I would like to indicate that Gallic acid has been reported to be a pro-oxidant or antioxidant but its pro-oxidant activity should be highlighted in the introduction

2) As indicated, a second method to measure cellular viability should be used in addition to the one reported by the authors. This is based on the reactivity of natural compounds, as it is the case of gallic acid to achieve the reduction of MTT and to produce an artifactual increase in the percentage of MTT reduced in respect to the controls [Hamada, et al. 2019, PharmaNutrition 7: 100140; M. Han et al. 2019 / Journal of Chinese Pharmaceutical Sciences 19 (2010) 195; van Tonder et al. 2015 BMC research notes 8: 47-47]

3) As previously indicated DCFH-DA is not specific for H2O2 and therefore the reported values can be overestimated. Superoxide anion radicals can greatly contribute to the oxidation of DCFH-DA. Authors should perform their experiments in the presence of quenchers or in the presence of enzymes consuming superoxide anion radicals like SOD or PEG-SOD if the measurements are performed in situ in cells. Otherwise, they should indicate that they are measuring something ambiguous, as it is ROS.

4) The authors answered my question but based on it, their dendrimer nanodevice will have a limited time to be used due to stability issues. They should indicate how stabilization was achieved, their half-life in water based buffers and how their methods and treatments.

5) The indicated sections “3.2.6. GA content determination” "3.2.6.1. Standard GA calibration curve by Folin Ciocalteu method” and "3.2.6.2. Estimation of GA concentration in GALD 7 (DL%)" are not proper for the Results Section and this is the reason to suggest their incorporation into the Materials and Methods section.

Author Response

REVIEWER #2

Although the authors answered all my questions I remain caution about potential pitfalls in this study.

1) To change the manuscript orientation is a suggestion based on the author's results but I agree with the author's response, the authors are the ones to decide how to orient the manuscript in one or another way. I would like to indicate that Gallic acid has been reported to be a pro-oxidant or antioxidant but its pro-oxidant activity should be highlighted in the introduction.

The authors agree with the Reviewer that the pro-oxidant activity of GA should be highlighted in the introduction, but they make kindly note that such statement was already present in the manuscript versions which the Reviewer has revised.

In addition, the authors notify that the pro-oxidant activity of GA was already highlighted also in the abstract. Please see the part of the manuscript highlighted in yellow and also reported below where you can read these statements.

ABSTRACT:

“Concerning this, although affected by several issues that limit their clinical application, antioxidant/pro-oxidant polyphenols, as gallic acid (GA), showed pro-oxidant anti-cancer effects and low toxicity for healthy cells, in several kind of tumors, not including NB.” (line 17-20, revised manuscript).

INTRODUCTION:

“Polyphenols, due to their antioxidant properties, have the possibility to act as preventive anti-cancer compounds [15], and thanks to their pro-oxidant effects, can work as mimics of chemotherapeutic drugs, inducing ROS-mediated cancer cell death [16].

It is the case of gallic acid (GA, Figure 1), which is the 3,4,5-tri-hydroxyls derivative of benzoic acid and represents one of the major phenolic acids present in various edible natural products, such as green tea, gallnuts, oak bark, apple-peels, grapes, strawberries, pineapples, bananas and many other fruits [17].” (lines 67-73).

“Inside plants, GA makes part of secondary metabolites involved in the formation of the galatotannin-hydrolysable tannins, but in biomedical sector, it has long attracted the interest of scientists for its ambivalent antioxidant/pro-oxidant behavior [18] and for its capacity in counteracting diseases correlated to OS, through its anti-bacterial, anti-viral, anti-inflammatory, anti-neurodegenerative and anticancer activities [15-17,19-29].” (lines 77-81).

2) As indicated, a second method to measure cellular viability should be used in addition to the one reported by the authors. This is based on the reactivity of natural compounds, as it is the case of gallic acid to achieve the reduction of MTT and to produce an artifactual increase in the percentage of MTT reduced in respect to the controls [Hamada, et al. 2019, PharmaNutrition 7: 100140; M. Han et al. 2019 / Journal of Chinese Pharmaceutical Sciences 19 (2010) 195; van Tonder et al. 2015 BMC research notes 8: 47-47]

The authors are forced to reiterate that a second method to measure cellular viability, in addition to the one reported in the manuscript, although it might seem a virtuosity, is superfluous and redundant, especially for a manuscript which aims at being published on Nanomaterials, i.e. a journal whose topics concern nanodevices and not on a journal particularly specific for cancer.

In addition, as reported in the previous rebuttal, the authors would be available to carry out further experiments to confirm the results reported in the manuscript, but due to the ongoing health emergency, the access to the laboratories is limited and they cannot carry out further assessments in the limited time of days at disposition for completing their revision.

Anyway, the authors reaffirm that, while on one hand the use of tetrazolium-based assays may give artifacts, on the other the high number of studies reporting the same essay leads us to believe, that the method herein used can be considered valid for evaluating cell viability. Please consider the following:

Duan et al., Cancer Letters 2010

Soman et al., J Immunol Methods 2009

Willems et al., J Vir Met 2011

Shi et al., BMC Cancer 2016

Gautam et al., Mol Cancer 2016

Gao et al., Cell Oncol 2018

Akter et al., Antioxidants 2019

Zhang et al., Cancer Cell Int 2019

Moreover, in order to validate their rebuttal, also in the sight of the further Reviewer comment concerning the reported issues about the use of MTT test specifically for GA, the authors make note that the MTT assay is used extensively as a method for assessing the cytotoxicity both of synthetic and/or natural compounds. In this context, just Nanomaterials has recently published a manuscript (Rosman R et al., Nanomaterials, 2018) in which the GA-induced cytotoxic action on cancer cells was assessed by MTT. Furthermore, the validity of this method is evidenced by many recent studies (Rouamba A et al., Antioxidants 2018; Li F et al., Molecules 2017; Heidarian E et al., Biomed & Pharamacoth 2016; Karimi A et al., Antiviral chem and chemotherapy 2020; Khorsandi K et al., Cancer Cell Int 2020).

Based on these considerations, we can reasonably confirm that the MTT assay is a valid method to evaluate the cytotoxic action of GA and therefore, we kindly ask that the Reviewer’s request be re-evaluated and ask the Reviewer, and the Academic Editor to understand the reasons that led us to not satisfy it.

3) As previously indicated DCFH-DA is not specific for H2O2 and therefore the reported values can be overestimated. Superoxide anion radicals can greatly contribute to the oxidation of DCFH-DA. Authors should perform their experiments in the presence of quenchers or in the presence of enzymes consuming superoxide anion radicals like SOD or PEG-SOD if the measurements are performed in situ in cells. Otherwise, they should indicate that they are measuring something ambiguous, as it is ROS.

In regard of the Reviewer request of the necessity to specify that something “ambiguous” as ROS have been measured and not H2O2, the authors make kindly note that in the version of the manuscript submitted after the first revision it was already indicated. Please see the heading of paragraph 2.12. previously "Detection of hydrogen peroxide (H2O2) production" that was changed in "2.12. Detection of reactive oxygen species (ROS) production" (line 390 of revised manuscript) and similarly the description of the method, where H2O2 was already replaced by ROS (lines 391-392).

Anyway, since the Reviewer reaffirms his position, similarly also the authors are forced to reiterate their one with the aim at validating the robustness of the method as follows.

According to Cossarizza A. et al. (Nature Protocols 2009), "DCFH-DA has a good specificity for H2O2, and it has been shown that the fluorescence of the product DCF appears to be mediated mainly by H2O2 ". In addition, the method that the authors use in this manuscript involves the lysis of cells in a solution of 90% DMSO and 10% PBS as suggested by Wang et al. (Free Radic. Res., 2009). In this study, Wang and colleagues report as follows: "Even though oxidation of intracellular DCFH to DCF may be affected by many factors and may not always be a measure of ROS formation, it nonetheless remains a useful and widely used probe of ROS production in cells. Cell lysis with 90% DMSO/10% PBS and the resulting homogenization and stability of the DCF fluorescence signal clearly results in a more precise and convenient measure of DCFH oxidation to DCF than other methods commonly used". Moreover, the reported method is widely used in order to evaluate ROS, but specifically H2O2 production (van de Wier et al., FEBS J 2013; Zhang et al., Cancer letters 2013; Martins et al., Autophagy 2018).

Finally, the author’s believe that the further Reviewer’s request to verify the amount of hydrogen peroxide instead of total ROS levels is not essential for the purpose of this study and for the target of Nanomaterials. In this regard, the authors reiterate that it is important to consider that only now they have started working in the laboratory after 3 months of lockdown and in a limited way and that at this moment they have some difficulties in carrying out experiments that, in addition they do not consider indispensable at this stage of the study.

4) The authors answered my question but based on it, their dendrimer nanodevice will have a limited time to be used due to stability issues. They should indicate how stabilization was achieved, their half-life in water based buffers and how their methods and treatments.

Concerning the Reviewer requests at point 4, additional details dealing with GALD stability, stabilization and its half-life in PBS solution have been included in the revised version of the manuscript. Please see lines 340, 683-687, 705-706, 714, 718-720, 721, 723, 727-728 and 736 of the revised manuscript. In addition, a new reference (Ref.81) have been included in the revised manuscript.

5) The indicated sections “3.2.6. GA content determination” "3.2.6.1. Standard GA calibration curve by Folin Ciocalteu method” and "3.2.6.2. Estimation of GA concentration in GALD 7 (DL%)" are not proper for the Results Section and this is the reason to suggest their incorporation into the Materials and Methods section.

Since the small three sentences paragraph 3.2.6. does not contain experimental data, it is hard for the authors consider it proper for Materials and Methods section and, although they are available to satisfy the Reviewer, they think that moving it could be pejorative for their manuscript.

In regard of sections 3.2.6.1. and 3.2.6.2., all the parts containing experimental data and not containing discussion have been moved to the Materials and Methods section, as requested. Please see modified parts at lines 263-269, 276-309, 317-338, 564-573, 578-585, 602-614, 616-625 and 636-644.

Round 3

Reviewer 2 Report

The authors made a great effort to answer the questions and modified the manuscript accordingly. Noteworthy, there is still a point that needs to be modified or corrected regarding section 3.3.

Reports indicate that gallic acid is able to reduce MTT  at very low concentrations [Journal of Chinese Pharmaceutical Sciences 19 (2010) 195–200].  Based on this ability, the reported increase in viability  (Figure 11) at high gallic acid concentrations is probably artifactual and probably are not reflecting the real cellular viability after cell exposure to gallic acid at high concentration. It does not make sense to have a viability of 150% (when cells are treated with 150 μM of gallic acid) over the control 100% (no gallic acid) unless the authors have an explanation for these results. The same type of problem might be behind ROS results, as shown in figure 10. Gallic acid induces a decrease of ROS production in respect to the control, at a high gallic acid concentration that might just reflect an artifactual protection against autoxidation or against DCF fluorescence photobleaching, as a simple explanation for these results. Moreover, if this supposed protection is real, it is contradictory to the pro-oxidant effects of gallic acid proposed and discussed in the rest of the paper.

I cannot accept the manuscript for publication in these conditions but I understand that Covid-19 situation made things complicated for all to go back to the labs.Since the high gallic concentrations (75 μM, 100 μM, and 150 μM)  section 3.3. are not used in the subsequent experiments with the dendrimers, they can be seen as controls for these experiments (sections 3.4 and 3.5). Therefore, I suggest eliminating the experiments performed at high gallic acid concentrations in ROS measurements and viability experiments (75 μM, 100 μM, and 150 μM) and to indicate that the tested concentrations of gallic acid (below 75 μM) are not toxic or induce changes in viability. Therefore, they can be used as controls for experiments performed with  nanodevices ( in which the maximum correlative used concentration of gallic acid was 21.2 μM)

Minor corrections: Please improve the quality of figures 8 and 9, normalize the axis tick marks, bounds and eliminate the crossed sections.

Author Response

REVIEWER #2

The authors made a great effort to answer the questions and modified the manuscript accordingly. Noteworthy, there is still a point that needs to be modified or corrected regarding section 3.3.

Reports indicate that gallic acid is able to reduce MTT  at very low concentrations [Journal of Chinese Pharmaceutical Sciences 19 (2010) 195–200].  Based on this ability, the reported increase in viability  (Figure 11) at high gallic acid concentrations is probably artifactual and probably are not reflecting the real cellular viability after cell exposure to gallic acid at high concentration. It does not make sense to have a viability of 150% (when cells are treated with 150 μM of gallic acid) over the control 100% (no gallic acid) unless the authors have an explanation for these results. The same type of problem might be behind ROS results, as shown in figure 10. Gallic acid induces a decrease of ROS production in respect to the control, at a high gallic acid concentration that might just reflect an artifactual protection against autoxidation or against DCF fluorescence photobleaching, as a simple explanation for these results. Moreover, if this supposed protection is real, it is contradictory to the pro-oxidant effects of gallic acid proposed and discussed in the rest of the paper.

I cannot accept the manuscript for publication in these conditions but I understand that Covid-19 situation made things complicated for all to go back to the labs. Since the high gallic concentrations (75 μM, 100 μM, and 150 μM)  section 3.3. are not used in the subsequent experiments with the dendrimers, they can be seen as controls for these experiments (sections 3.4 and 3.5). Therefore, I suggest eliminating the experiments performed at high gallic acid concentrations in ROS measurements and viability experiments (75 μM, 100 μM, and 150 μM) and to indicate that the tested concentrations of gallic acid (below 75 μM) are not toxic or induce changes in viability. Therefore, they can be used as controls for experiments performed with  nanodevices ( in which the maximum correlative used concentration of gallic acid was 21.2 μM)

The authors, reading carefully the Reviewer's comments, could not manage to understand why he spoke of ROS reduction and increase of cell viability at the aforementioned high GA concentrations used in the time course dose-dependent experiments reported in the manuscript, which, in fact, were given in contrast with the proposed ROS-mediated cytotoxic activity praised along all the work.

Consequently, the authors went to see the revised versions R1 and R2 of the original work submitted to Nanomaterials and realized that, in both R1 and R2, the Figures 10 and 11, showing the GA activities dependent on the dosage and time, had been reversed in respect of the manuscript originally sent to Nanomaterials.

As the same Reviewer will remember, in the first revision phase, the authors were asked to insert the titles of the y-axes in Figures 10, 11, 12 and 13. In doing so, the authors inverted Figure 10 with 11, thus reversing the findings, that seemed to be artifacts or anyway meaningless and in disagreement with what was said in the text, as the Reviewer rightly commented. We ask the Reviewer to go and check the first version of our work that he received (which we attach), where he can check and verify what has been herein said.

In this new revised version of the manuscript, apologizing to the Reviewer for such distraction which also led to misunderstandings between the parts, the correct Figures 10 and 11 have been inserted in paragraph 3.3. The Reviewer will be able to ascertain that in this way the reported data no longer seem to be artifacts and agree with what was discussed in the text.

Minor corrections: Please improve the quality of figures 8 and 9, normalize the axis tick marks, bounds and eliminate the crossed sections. Although the authors answered all my questions I remain caution about potential pitfalls in this study.

As suggested, the quality of Figures 8 and 9 has been improved by normalizing the axis tick marks, bounds and eliminating the crossed sections.

The authors are confident that, having solved the question of Figures 10 and 11 inversion, and having explained the occurred misunderstandings, the possible pitfalls that could cause concern in the Reviewer, may have been nullified.

Sincerely,

Alfei and co-authors                                    
